# Potential Range Shift of Snow Leopard in Future Climate Change Scenarios

Xinhai Li [1,2,*], Liming Ma [1,3], Dazhi Hu [4], Duifang Ma [4], Renqiang Li [5], Yuehua Sun [1,2,*] and Erhu Gao [6,*]

1 Key Laboratory of Animal Ecology and Conservation Biology, Institute of Zoology, Chinese Academy of Sciences, Beijing 100101, China; maliming717@163.com

2 College of Life Sciences, University of Chinese Academy of Sciences, Beijing 100049, China

3 School of Life Sciences, Hebei University, Baoding 071002, China

4 Qilianshan National Park Administration Bureau Zhangye Branch, Zhangye 734000, China; hdz1@163.com (D.H.); mdfgogo@163.com (D.M.)

5 Institute of Geographic Sciences and Natural Resources Research, Chinese Academy of Sciences, Beijing 100101, China; renqiangli@igsnrr.ac.cn

6 Academy of Inventory and Planning, National Forestry and Grassland Administration, Beijing 100714, China

* Correspondence: lixh@ioz.ac.cn (X.L.); sunyh@ioz.ac.cn (Y.S.); gaoerhu2022@126.com (E.G.); Tel.: +86-10-64807898 (X.L.); +86-10-64807132 (Y.S.); +86-10-84238082 (E.G.)

**Abstract:** The snow leopard (*Panthera uncia*) lives in alpine ecosystems in Central Asia, where it could face intensive climate change and is thus a major conservation concern. We compiled a dataset of 406 GPS-located occurrences based on field surveys, literature, and the GBIF database. We used Random Forest to build different species distribution models with a maximum of 27 explanatory variables, including climatic, topographical, and human impact variables, to predict potential distribution for the snow leopard and make climate change projections. We estimated the potential range shifts of the snow leopard under two global climate models for different representative concentration pathways for 2050 and 2070. We found the distribution center of the snow leopard may move northwest by about 200 km and may move upward in elevation by about 100 m by 2070. Unlike previous studies on the range shifts of the snow leopard, we highlighted that upward rather than northward range shifts are the main pathways for the snow leopard in the changing climate, since the landform of their habitat allows an upward shift, whereas mountains and valleys would block northward movement. Conservation of the snow leopard should therefore prioritize protecting its current habitat over making movement corridors.

**Keywords:** alpine ecosystem; conservation planning; global warming; model selection; species distribution model; range shift; representative concentration pathways (RCPs)

## 1. Introduction

The snow leopard (*Panthera uncia*) has lived on the earth for millions of years [1], survived historical climate fluctuations [2], and currently maintains a population of about 4000 individuals [3]. Its population trend is stable in some places and decreasing in others [4], whereas its status in the International Union for Conservation of Nature (IUCN) Red List was upgraded from Endangered to Vulnerable [3], based on the long-standing and widely accepted assessment of its global population. The large cat mostly inhabits the alpine zone at 3000–5000 m between the snow line and the tree line [2,5,6], a climate-sensitive area expected to experience rapid global warming in the near future [7]. How climate change would affect snow leopard populations on a human-dominated planet is a major issue for their conservation [2,5,8].

Several studies have evaluated the effects of climate change on the snow leopard. Li et al. built a habitat map of the snow leopard from the last glacial maximum to 2070, which predicted the climate refugia for this species [8]. Riordan et al. assessed potential

connectivity across the range of the snow leopard and suggested corridors for individual dispersal between populations under the pressure of climate change [9]. However, most studies were conducted at local scales, e.g., [5,10–12], rather than at a range-wide scale, or using inaccurate occurrences (uncertainty margin of about 10 km) based on the names of villages or towns rather than GPS locations, e.g., [4,8]. The latest range-wide research was that of Li et al., which defined the conservation priorities for the snow leopard using over 6000 occurrence records [4]. With the accumulation of snow leopard occurrences from field surveys in recent years, a range scale analysis of the potential impacts of climate change on the snow leopard based on accurate GPS-derived records is thus now feasible.

Species distribution models (SDMs) are appropriate tools for assessing the effects of climate change on species [13]. SDMs may utilize different algorithms, including regressions, classifications, or complex models [14], and can relate species occurrences to environmental predictors and generate habitat suitability maps [15,16]. Some researchers have applied SDMs to study the habitat use of the snow leopard in protected areas in China [17,18], Russia [19], India [20], Nepal [21], Pakistan [22], and Kazakhstan [23]. However, a range-wide study on the species–environment relationship for the snow leopard is expected to show the overall pattern of distribution and habitat use.

In the past decade, numerous surveys have been conducted for the snow leopard; in particular, camera trapping has obtained a large number of accurate locations of this species in many countries and has provided an opportunity for further analysis [24,25]. In this paper, we aim to build a range-wide SDM for the snow leopard using recently obtained high-quality GPS locations, project its potential future range shift, and compare the importance of climate variables with other factors such as elevation and human influence in regard to determining the snow leopard's range at a broad scale. Our study will provide useful information for coordinating cross-border cooperation for snow leopard conservation in the 12 countries where the species lives.

## 2. Materials and Methods

### 2.1. Data

The snow leopard occurrence data used in this study were from field surveys, literature, and the Global Biodiversity Information Facility (GBIF) database (Table 1). Most of the occurrences were derived from camera trap records; others were from feces, or footprints marked during line transect surveys. Because the snow leopard lives in steep mountains, tens or hundreds of meters of location error could refer to different habitat conditions. We only selected accurate GPS-located occurrences (with error < 15 m), and points within a 5 km radius of another point were removed, in order to reduce spatial pseudoreplication. In total, 352 occurrences were removed from 758 records (Figure S1). Snow leopard individuals usually have stable home ranges, and their sizes vary in different regions, such as 11.2–22.5 km$^2$ [26], 11.5–100 km$^2$ [27], and 16.7–25 km$^2$ [28]. As such, occurrences 2 to 3 km apart may belong to the same individual, whereas occurrences 5 km apart are likely to belong to different individuals. Although nearly half of the occurrences were removed because they were close to each other, the presence of snow leopards still looked cluster-distributed (Figure 1). Such an awkward situation reflects that the survey effort for snow leopards is far from sufficient. In this study, the regions with a high density of occurrences were three national parks (i.e., Sanjiangyuan, Qilianshan, and Wolong) in China (Table 1), where intensive surveys were carried out in good habitats with a high density of blue sheep (*Pseudois nayaur*) and a low level of human activity. We recognized that this could lead to bias towards these three parks in the species distribution models, and we paid special attention to its consequences.

**Table 1.** Occurrences (selected from 758 original records) for snow leopards used in the study.

| Number of Occurrences | Type * | Region | Source |
|---|---|---|---|
| 43 | Direct observation or camera trapping | All habitats ** | GBIF [29] |
| 164 | Survey or camera trapping | China | [30] |
| 47 | Feces collection or camera trapping | Sanjiangyuan *** | [31] |
| 15 | Survey | Sanjiangyuan *** | Unpublished data |
| 53 | Camera trapping | Qilianshan *** | Unpublished data |
| 27 | Survey | Qilianshan *** | Unpublished data |
| 19 | Camera trapping | Wolong *** | [32] |
| 30 | Camera trapping | India | [20] |
| 8 | Direct observation | India | [20] |
| 406 | Total | / | / |

* "Survey" refers to a line transect survey. ** Including occurrences in India (20 points), Mongolia (9 points), Nepal (5 points), China (5 points), Kyrgyzstan (2 Points), Afghanistan (1 point), and Russia (1 point). *** The regions are national parks in China.

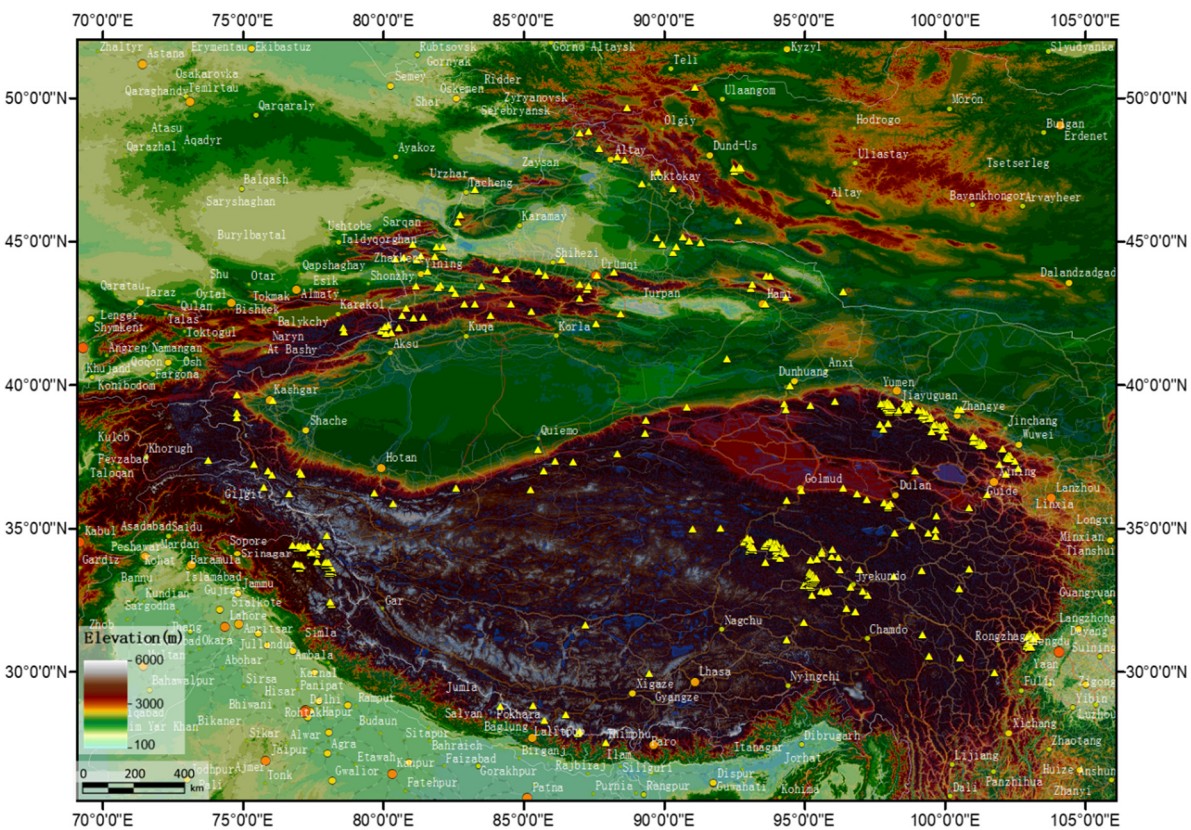

**Figure 1.** Snow leopard occurrences (yellow triangles). The brown dots are cities. The white lines are country borders. The background is an elevation map.

All occurrences used for SDMs were from recent surveys conducted after 2010, and more than 60% of the records were based on surveys from the past five years.

We used the 19 bioclimate variables (30 arc-second) from the WorldClim 2.0 dataset (https://worldclim.com/, accessed on 19 August 2021) [33] for current and future climate conditions. The new Shared Socioeconomic Pathways (SSPs) had been proposed with five pathways, taking into account socioeconomic factors such as population, economic growth, urbanization, education, and the rate of technological development [34], and had been used as important inputs for the latest climate models [35]. However, the data supporting these scenarios are still not publicly available. In this study, future climate data were derived from global climate models (GCMs), which are based on the general principles of fluid

dynamics and thermodynamics, describing the dynamics of the atmosphere and ocean in an explicit way [36]. The future data we used were from the CMIP5 (Coupled Model Intercomparison Project Phase 5) downscaled future climate projections [37]. Downscaling and calibration were performed with WorldClim v1.4 as baseline climate [33]. The CMIP6 data at 30-s spatial resolution are still not available. We selected two GCMs, CCSM4 and IPSL-CM5A-LR, because they are moderate models with temperature and precipitation neither too high nor too low [38], and their data are available for all the four representative concentration pathways (RCPs 2.6, 4.5, 6.0 and 8.5) for both time periods, 2050 (average for 2041–2060) and 2070 (average for 2061–2080).

The elevation data (30 arc-second) were originally acquired by spaceborne shuttle radar [39] and downloaded from the website of the United States Geological Survey (USGS) [40]. The human footprint index (v2, 1995–2004) is a raster dataset with a resolution of one kilometer, created from nine global data layers covering human population density, human land use, human access (coastlines, roads, railroads, and navigable rivers), and infrastructure (built-up areas, nighttime lights, and land use/land cover) [41]. The data for solar radiation, wind speed, and water vapor pressure were downloaded from WorldClim [42]; data for each of the three variables were available for each month, and we selected values in January and July for this study.

All environmental data were raster layers with a resolution of 30 arc-seconds (around 1 km at equator), for a total of 27 variables.

### 2.2. Models

We used environmental variables to explain the presence or absence of the snow leopard using the model $\text{logit}(p) = f(x1, x2, \ldots xk)$, where $p$ is the probability of the presence of snow leopard at a location, and $x$ are environmental variables.

We applied Random Forest [43] for SDMs. Random Forest is a machine-learning algorithm and is one of the best performing SDM approaches [44,45]. It is especially good for handling high-dimensional data (multiple correlated explanatory variables) and complex relationships such as interaction and high-order effects [14,46,47]. We used the R package randomForest [48] for the analysis. We set the argument ntree (number of trees) to be 1000, and left the others undefined (default values were used). The default settings of random forest usually provide good results, unlike other SDM algorithms such as Maxent, support vector machine, or artificial neural networks.

For the current distribution of snow leopards, we calibrated the model by considering two aspects: (1) different numbers of pseudoabsences and (2) different combinations of explanatory variables. We tried evenly distributed $50 \times 50 = 2500$ and $22 \times 22 = 484$ pseudoabsence points, respectively. The model with 2500 absence points, six times higher than the number of present points (406), was a low-prevalence model. The model with 484 absence points was a balanced-prevalence model. We used the 19 climate variables in the Worldclim dataset, elevation, human footprint index, and January and July values for solar radiation (kJ m$^{-2}$ day$^{-1}$), wind speed (m s$^{-1}$), and water vapor pressure (kPa) as explanatory variables for a total number of 27 variables. We also built a model that only used the 19 climate variables and compared model performance to explore the contribution of nonclimate variables to the model. To deal with the potential multicollinearity of climate variables, we used the vif() function in the R package car to calculate variance inflation factors (VIFs) and selected variables with VIF values less than five. We named the model with 27 explanatory variables the full model, the model with 19 climate variables the climate model, and the model with independent climate variables (less than 19 variables) the simple climate model.

There were 77 presence and absence points with null values for the human footprint index. We used na.roughfix() function in R package randomForest to replace the null values with the median of the human footprint index.

We ranked the importance for all 27 variables following the Random Forest evaluation. Random Forest provides two indices for variable importance, mean decrease accuracy and

mean decrease Gini. The former is for regression, measuring the decreased prediction accuracy after removing a variable; the latter is for classification, quantifying the decreased Gini purity at nodes after removing a variable [43].

For the future distribution of snow leopards, we first built a model using 19 climate variables for current conditions, and then applied the model to predict the habitat suitability of snow leopards in the future using GCMs CCSM4 and IPSL-CM5A-LR for RCPs 2.6, 4.5, 6.0, and 8.5 for 2050 and 2070, respectively. The four representative concentration pathways from low to high corresponded to different global warming degrees from low to high.

## 3. Results

### 3.1. Model Calibration

We selected seven independent variables from the 19 climate variables, which had VIF values less than five (Table S1). The variables were: annual mean temperature (bio_1); mean diurnal range (mean of monthly (max temp − min temp)) (bio_2); temperature seasonality (standard deviation *100) (bio_3); annual precipitation (bio_12); precipitation of driest month (bio_14); precipitation seasonality (coefficient of variation) (bio_15); and precipitation of coldest quarter (bio_19).

The models were run under six settings: two pseudoabsence levels (2500 vs. 484) and three explanatory variable combinations (27 variables in the full model vs. 19 variables in the climate model vs. 7 variables in the simple climate model), i.e., low-prevalence and balanced-prevalence data for the full model, climate model, and simple climate model, respectively. We listed the model performance indices based on 1000 trees from the Random Forest (Table 2). We found that the overall accuracy ((true presence + true absence)/(occurrences + pseudoabsence)) of all the models was above 90%. The low-prevalence models with 2500 pseudoabsence points outperformed the balanced-prevalence models. They projected a potential habitat for the snow leopard with a lower presence probability and had much less commission errors (1-specificity, 3.2% vs. 8.9%) in the climate models (Table 2 and Figure 2). The full models took into account elevation, human influence, solar radiation, wind speed, and water vapor pressure, yet these variables, compared with the climate model, only slightly improved the model performance, decreasing the error rate by 0.3% for the low-prevalence model and 0.5% for the balanced-prevalence model (Table 2). The simple climate model which used 7 independent climate variables had a similar performance to the climate model with 19 variables (Table 2).

**Table 2.** The performance of species distribution models for snow leopards under six settings: low-prevalence (2500 pseudoabsence points) and balanced-prevalence (484 pseudoabsence points) data for full model (27 variables), climate model (19 variables), and simple climate model (7 variables), respectively. The mean and standard deviation (SD) of the indices were based on 1000 trees from Random forest.

| | No. Presence Points | No. Pseudoab­sence Points | Accuracy (%) ±SD | Sensitivity ±SD | Specificity ±SD |
|---|---|---|---|---|---|
| Full model | 406 | 484 | 91.2 ± 0.88 | 90.5 ± 1.67 | 91.8 ± 1.35 |
| Full model | 406 | 2500 | 95.1 ± 0.40 | 83.2 ± 2.02 | 97.0 ± 0.42 |
| Climate model | 406 | 484 | 90.7 ± 0.86 | 90.1 ± 1.62 | 91.1 ± 1.47 |
| Climate model | 406 | 2500 | 94.8 ± 0.39 | 82.6 ± 2.04 | 96.8 ± 0.43 |
| Simple climate model | 406 | 484 | 90.6 ± 0.92 | 89.9 ± 1.73 | 91.1 ± 1.40 |

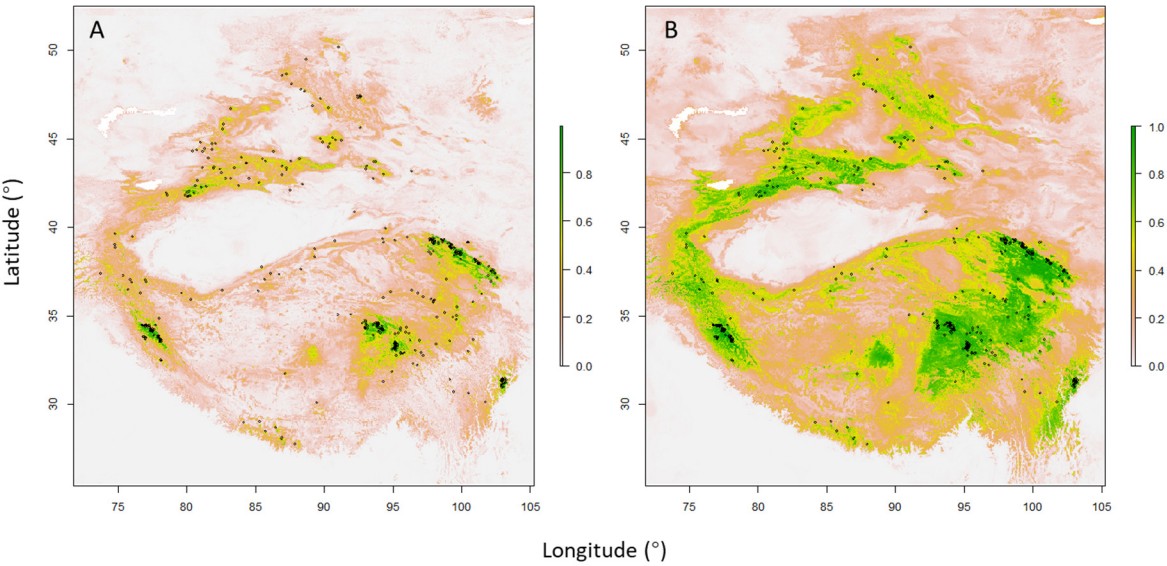

**Figure 2.** The predicted probability of snow leopard presence under current climate conditions for the low-prevalence model (**A**) and balanced-prevalence model (**B**). The black dots are snow leopard occurrences.

### 3.2. Variable Importance

The 27 explanatory variables in the full model had different influences on the snow leopard habitat modelling. Precipitation seasonality (bio_15), solar radiation in January, elevation, human footprint index, solar radiation in July, and water vapor pressure in July were the six most important variables as measured by mean decrease accuracy. Solar radiation in January, elevation, water vapor pressure in July, precipitation seasonality, temperature seasonality (bio_4), and mean temperature of driest quarter (bio_9) were the six most important variables as measured by mean decrease Gini (Figure S2). The important variables included temperature and precipitation variables, and the variances of these two factors (e.g., precipitation seasonality and temperature seasonality) were more important than their mean values (e.g., mean temperature and total precipitation). Other variables such as elevation, human footprint index, solar radiation, and water vapor pressure also appeared to be important.

However, the variables that appeared to be the most important in Random Forest were not definitive, because many variables were highly correlated. We found strong positive correlations between temperature seasonality (bio_4) and temperature annual range (bio_7), max temperature of warmest month (bio_5) and mean temperature of warmest quarter (bio_10), and annual mean temperature (bio_1) and water vapor pressure; we found strong negative correlations between isothermality (bio_3) and temperature seasonality (bio_4), and elevation and mean temperature (bio_5 and bio_10) (Figure S3).

### 3.3. Future Range Shift

The climate model with low-prevalence data had a lower error rate (Table 2) and gave a more conservative prediction for snow leopard presence (Figure 2), so we selected this model to project the current suitable habitat (Figure 3) and any potential range shift of snow leopards in the future (Figure 4). The difference between the GCMs CCSM4 and IPSL-CM5A-LR were small, and the difference among the four representative concentration pathways were substantial (Figures 3 and 4). The distribution center (mean latitude and longitude of all quadrats in the study area, weighted by the probability of presence in Figure 2A) of snow leopards would move northwest by about 200 km in 2070 (Figure 5). The mean elevation of snow leopard distribution for the eight climate change scenarios would increase by about 100 m in 2070 (Figure 6). The range shift of snow leopards in

2050 would be less than that in 2070. The density curve of the elevation of the evenly distributed pseudoabsence points had a peak at 4500–5400 m (Figure 6), meaning the area at the elevation range of 4500–5400 m is large. A potential habitat at higher elevations is available for the snow leopard.

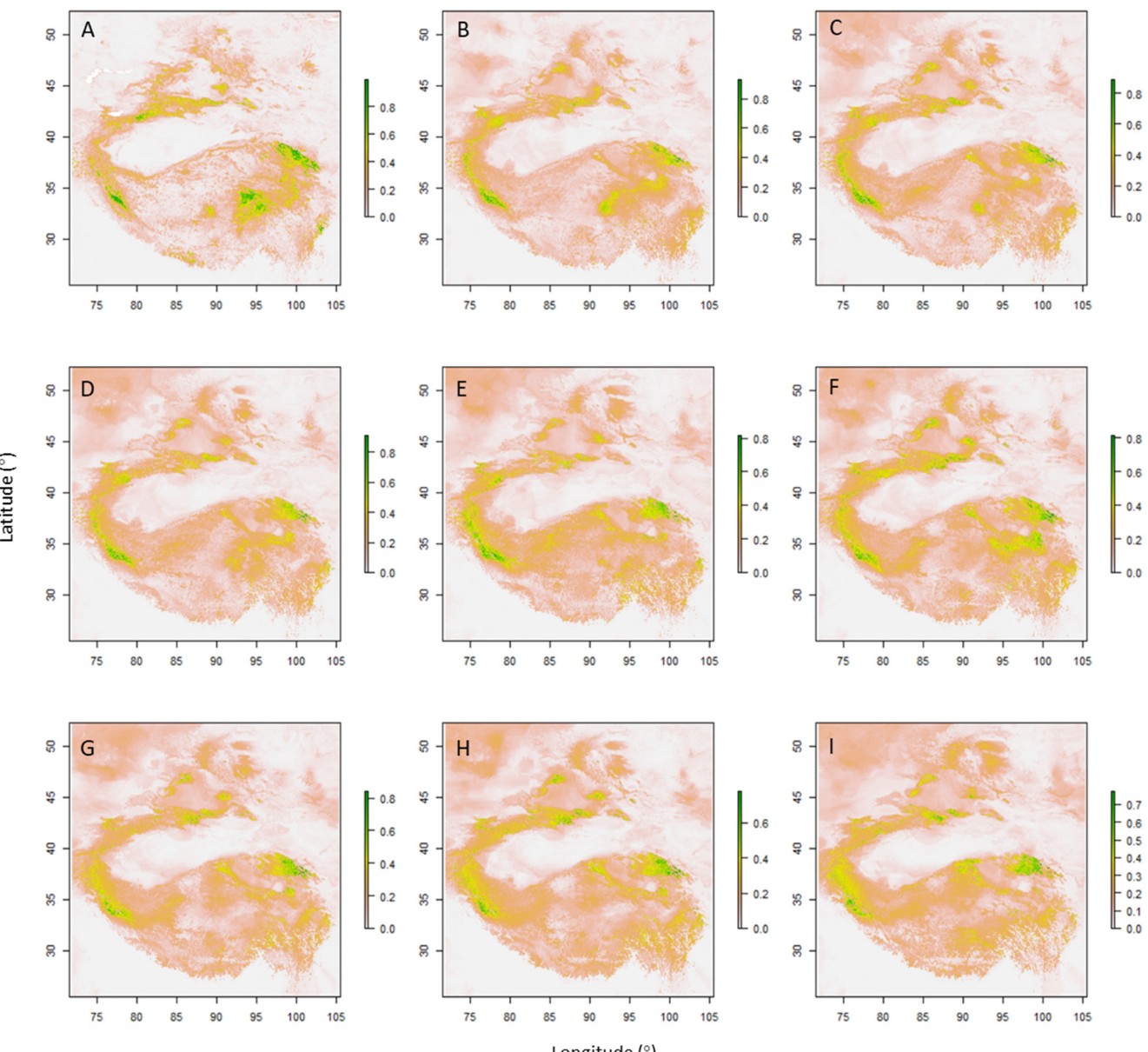

**Figure 3.** The predicted probability of snow leopard presence under current climate conditions (**A**), and under future climate change scenarios: GCMs CCSM4 for RCPs 2.6 (**B**), 4.5 (**C**), 6.0 (**D**), and 8.5 (**E**) in 2070, and IPSL-CM5A-LR for RCPs 2.6 (**F**), 4.5 (**G**), 6.0 (**H**), and 8.5 (**I**) in 2070.

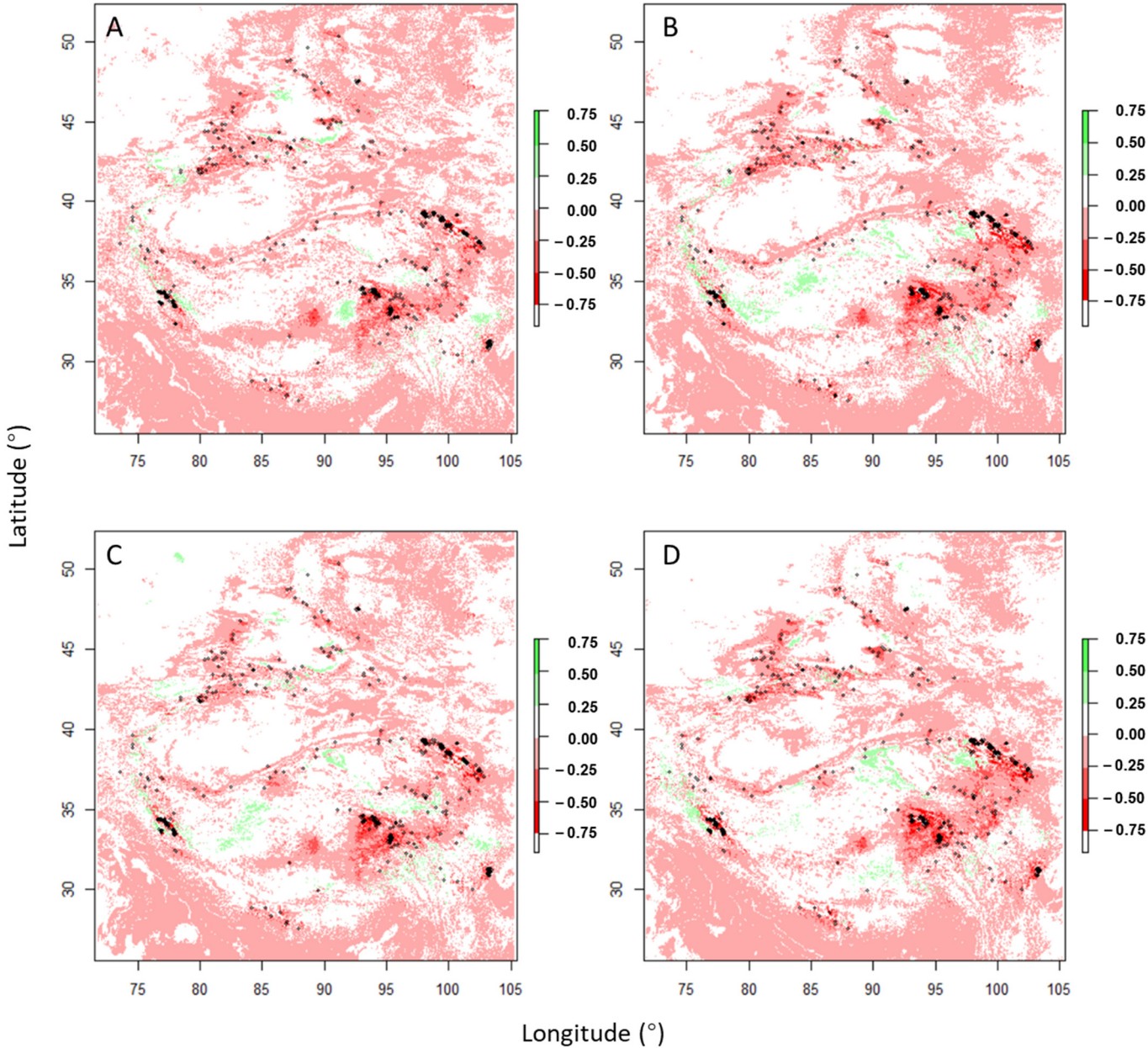

**Figure 4.** The predicted potential range shift of snow leopards under future climate change scenarios: GCMs CCSM4 for RCPs 2.6 (**A**) and 8.5 (**B**) in 2070, and IPSL-CM5A-LR for RCPs 2.6 (**C**) and 8.5 (**D**) in 2070. The values of range shifts were calculated by the probability of presence in the future minus the probability of presence at current climate conditions. The positive values mean more suitable habitat, and the negative values mean less suitable habitat in the future.

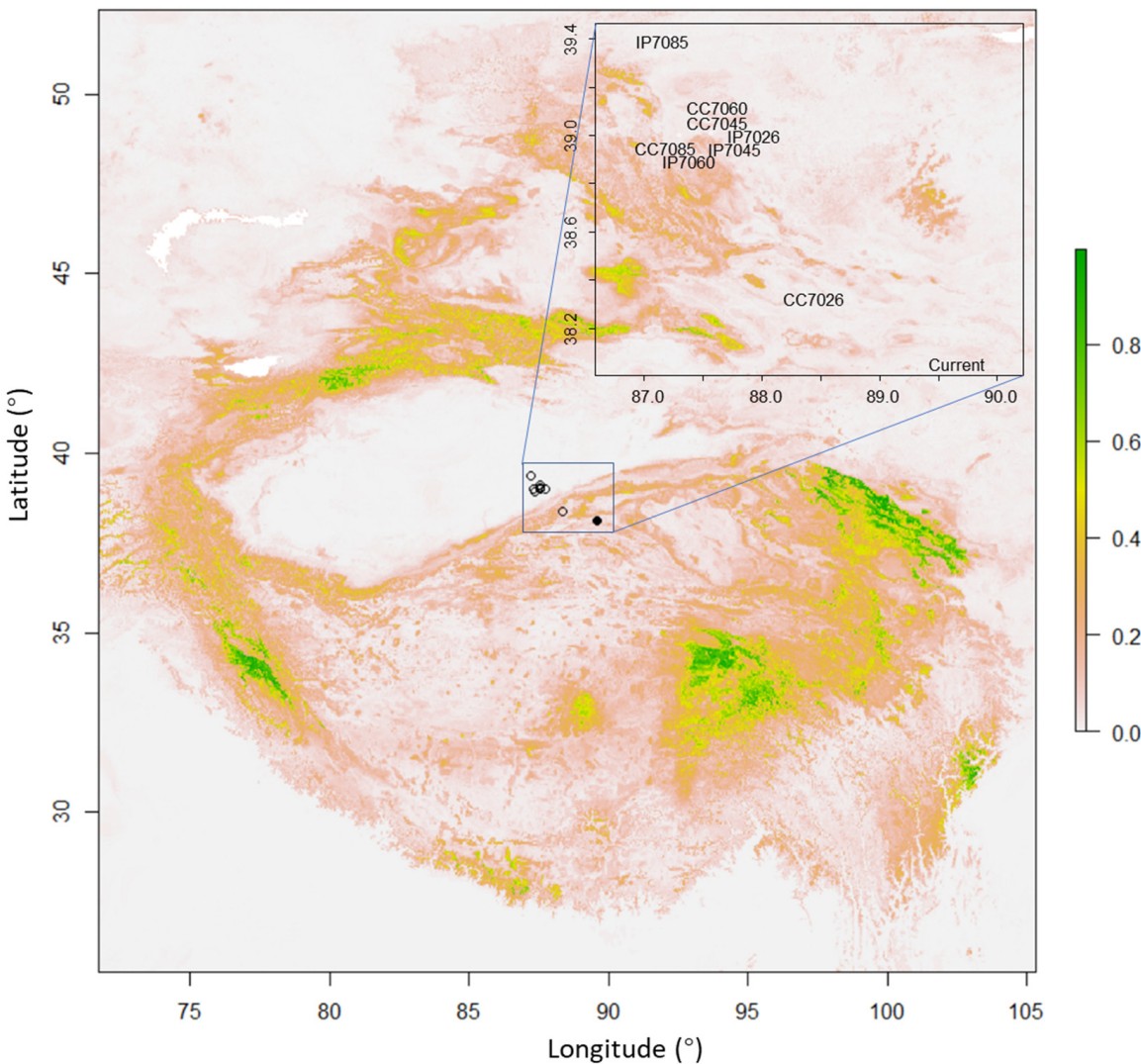

**Figure 5.** The predicted distribution centers of snow leopards under current climate conditions (solid dot) and future climate change scenarios (open circles). Northward range shifts in the future are demonstrated. The background is predicted probability of snow leopard presence under current climate conditions. CC7026, CC7045, CC7060, and CC7085 refer to GCM CCSM4 for RCPs 2.6, 4.5, 6.0, and 8.5 for 2070, respectively. IP7026, IP7045, IP7060, and IP7085 refer to GCM IPSL-CM5A-LR for RCPs 2.6, 4.5, 6.0, and 8.5 for 2070, respectively.

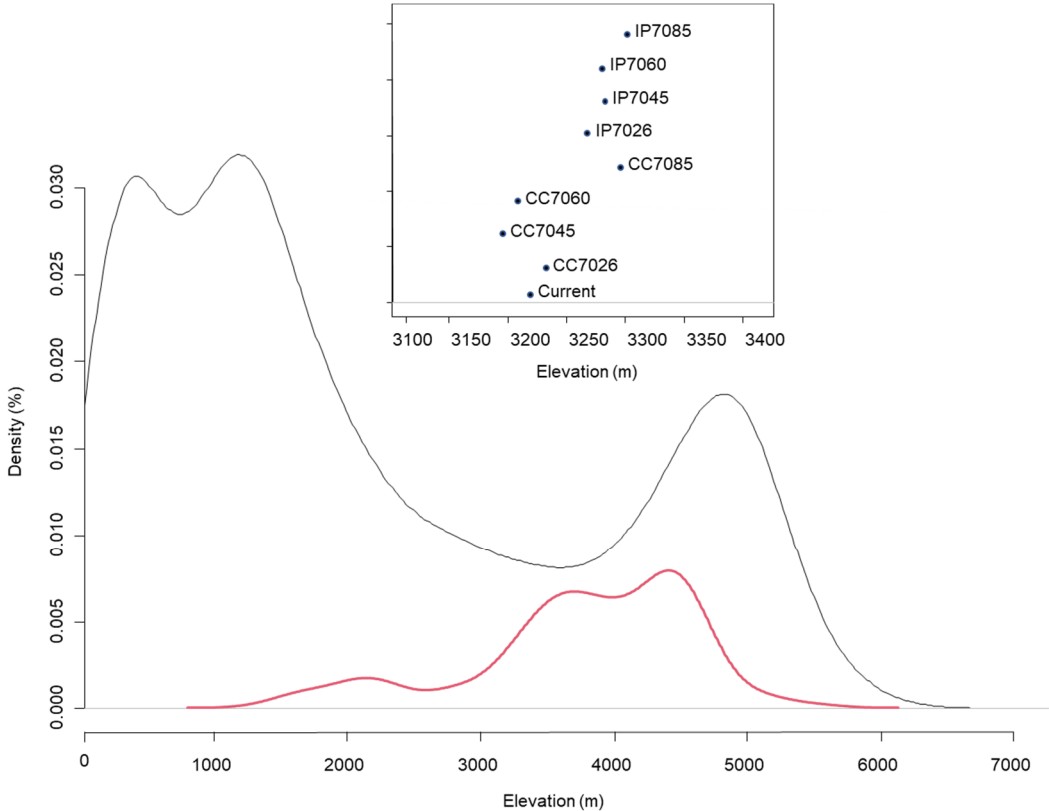

**Figure 6.** The predicted elevation of snow leopards under current climate conditions and future climate change scenarios. The black curve is the probability density of elevation of 2500 pseudoabsence points. The red curve is the probability density of elevation of 406 snow leopard occurrences. The upper panel shows the mean elevation of the predicted snow leopard distribution for certain climate change scenarios. CC7026, CC7045, CC7060, and CC7085 refer to GCM CCSM4 for RCPs 2.6, 4.5, 6.0, and 8.5 in 2070, respectively. IP7026, IP7045, IP7060, and IP7085 refer to GCM IPSL-CM5A-LR for RCPs 2.6, 4.5, 6.0, and 8.5 in 2070, respectively.

## 4. Discussion

While there are numerous studies on snow leopard habitats, range-wide habitat analyses of the current and future potential climate conditions of this species are still rare (except [4,8]). The major difficulty in such analyses is the lack of occurrence data, since surveys at high elevations are difficult and expensive. In the last decade, many new field survey projects have been carried out and more data are now available [30]. In particular, new and widely used camera-trapping surveys have provided a large amount of high-quality data for snow leopard occurrences e.g., [49–52]. Here, we showed that snow leopard occurrence data compiled from published literature, camera-trap surveys, and line transect surveys can be applied using Random Forest to effectively model snow leopard habitats and project the suitable habitats of snow leopards under future climate scenarios.

Animal habitat selection involves numerous variables with complex high-order and interaction effects, and linear models such as logistic regression are usually unable to fit these data well [53]. Random Forest and Maxent are often claimed to be the two top algorithms for species distribution modelling [14,44]. In order to better control the number of absence data included, we applied Random Forest to analyze snow leopard habitats, as Maxent selects pseudoabsence points automatically by default. In particular, the explanatory variables in the full model and climate model were highly correlated, and Random Forest is especially effective for such a situation, as it can provide accurate predictions without removing correlated variables [54,55].

However, Random Forest is likely to overfit models to a dataset. In our case, we took advantage of this tendency and developed a low-prevalence model, which was likely to predict the presence of snow leopards at well-surveyed areas and predict absences in unsurveyed areas. Such a model has a very low commission error yet a high omission error (Table 2). The omission is caused by survey gaps, as expected. As new survey data accumulate and are added to our modeling framework, the omission error will drop gradually, further improving the model.

The snow leopard is a habitat specialist with specific environmental requirements [17], such that species distribution models work well for snow leopards. We compared the contribution of 27 potential explanatory variables and found that many nonclimate variables (such as elevation, human footprint index, solar radiation, and water vapor pressure) played important roles (Figure S1). However, the nonclimate variables and climate variables were correlated (Figure S3). As a result, the climate models (that only used 19 climate variables) also demonstrated a good performance (Table 2). We used the climate model to project the potential range shift of the snow leopard.

Climate change is likely to cause an upward shift of the snow line and tree line, permafrost degradation, glacier retreat, and alteration to the distribution of snow leopards. Our SDMs projected that the snow leopard would have an upward range shift of about 100 m (Figure 6), and a northwestward range shift of about 200 km (Figure 5) in 2070. Li et al. predicted a dramatic northward range shift of the snow leopard in 2070 (RCP8.5) [8], which is different to our results. We think a likely reason for this difference is that the two studies used different occurrence datasets. As mentioned above, the current occurrence datasets are biased samples of the snow leopard population, since a large proportion of the snow leopard habitat has never been surveyed. On the other hand, surveyors usually set cameras in more accessible areas, so higher and remoter areas still largely remain unsurveyed. Besides, researchers have generally selected habitats to survey that are known to be frequently used by snow leopards, and marginal habitats are often ignored [56]. At present, there is still a large need for additional snow leopard investigation and data collection [30].

We argue that an upward range shift, not a northward range shift, is the main pathway for snow leopards in the changing climate (Figure 4). Given the current distribution of the snow leopard, there are spaces available for upward range shifts (Figure 6). We predicted that the snow leopard would have a slightly larger suitable habitat in the future (Figure 3), which is consistent with other studies, e.g., [2]. Li et al. [8] projected that the central part of the current snow leopard habitat would remain climatically suitable through the late 21st century, supporting viable populations, and would function as refugia for snow leopards. Given the present stable status and future viability of their habitat, we believe the conservation priority regarding snow leopards is to protect their current habitat rather than to create movement corridors.

The snow leopard is a keystone species for the alpine ecosystem in Central Asia and its status attracts attention, with numerous projects having been conducted to investigate and conserve the species [57,58]. In spite of the long history of human–snow leopard conflict [59–62], the large cat is one of the species with the least human disturbance (Figure S4) across its range, which has only declined in a few regions, such as the eastern part of its range within the Giant Panda National Park [63] and the Tomur National Nature Reserve of Xinjiang, Northwest China [64]. The snow leopard retains a healthy population in a broad area, and its conservation status was upgraded from Endangered to Vulnerable in 2017 [3].

However, as we have shown here, climate change is a potential threat to the snow leopard [2,5], though we predicted that the threat is minor. All of the 12 countries which the snow leopard inhabits should strengthen cross-border cooperation, closely monitor the snow leopard population status, and develop appropriate conservation action plans to secure the long-term survival of the snow leopard.



**Supplementary Materials:** The following are available online at https://www.mdpi.com/article/10.3390/su14031115/s1, Table S1: Variance inflation factors (VIFs) for the seven variables among the 19 climate variables, Figure S1: Removing snow leopard occurrences within five km radius at a camera-trap zone in Qilianshan National Park. The red triangles are removed occurrences and the green triangles are retained occurrences. The background is an elevation map with one km resolution, Figure S2: The importance of environmental variables in the species distribution model for snow leopard quantified by Random Forest. Variables bio_1 to bio_19 are 19 climate variables in Worldclim database. Variable elev is elevation; footprint is human footprint index; solar1 and solar7 are solar radiation in January and July, respectively; vapor1 and vapor7 are water vapor pressure in January and July, respectively; and wind1 and wind7 are wind speed in January and July, respectively, Figure S3: The Pearson correlation map for 27 environmental variables at 406 snow leopard occurrences. Variables bio_1 to bio_19 are 19 climate variables in Worldclim database. Variable elev is elevation; footprint is human footprint index; solar1 and solar7 are solar radiation in January and July, respectively; vapor1 and vapor7 are water vapor pressure in January and July, respectively; and wind1 and wind7 are wind speed in January and July, respectively. Figure S4. Snow leopard occurrences and human footprint index (v2, 1995–2004, Wildlife Conservation Society 2005).

**Author Contributions:** Conceptualization, X.L. and E.G.; methodology, X.L.; validation, D.H., D.M., and R.L.; formal analysis, X.L.; investigation, L.M., D.H. and D.M.; data, X.L. and R.L.; writing—original draft preparation, X.L.; writing—review and editing, X.L., Y.S. and E.G.; visualization, X.L.; supervision, Y.S.; project administration, X.L.; funding acquisition, X.L. All authors have read and agreed to the published version of the manuscript.

**Funding:** This work was supported by the National Natural Science Foundation of China (No. 31970432 and 31772479), the Second Tibetan Plateau Scientific Expedition and Research Program (STEP, Grant No. 2019QZKK0501), the Third Xinjiang Scientific Expedition Project (Grant No. 2021XJKK1302), the Second National Wildlife Survey Project for Terrestrial Animals, and the National Key Program of Research and Development, Ministry of Science and Technology (2016YFC0503200).

**Institutional Review Board Statement:** Ethical review and approval were waived for this study.

**Informed Consent Statement:** Not applicable.

**Data Availability Statement:** The occurrences of snow leopards are available upon request from the corresponding author, since the data are involved in a multiagent cooperation and cannot be shared publicly at present.

**Acknowledgments:** We are grateful to Mary Blair and Ming Xu for providing valuable comments on this manuscript.

**Conflicts of Interest:** The authors declare no conflict of interest.

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
