# Peer review of "Potential Range Shift of Snow Leopard in Future Climate Change Scenarios"

_sustainability, doi:10.3390/su14031115_

Round 1

Reviewer 1 Report

This article reported the potential distribution shift for the snow leopard in the future with variate climate change scenarios.  Several comments as following:

  1. The conclusion in Line 306-307 "Conservation priority of snow leopard is to protect 306 current habitat rather than making movement corridors." require more supporting materials.
  2. In Line 314 stated "climate change is a potential threat to the snow leopard", but also said in Line 304-305 "We predicted the snow leopard would have slightly larger suitable habitat in the future." This needs to be clarified.
  3. The discussion on using Random Forest is better than previous modeling reported using other models requires more supports in the discussion.
  4. The resolution of all color maps need to be improved.  Also, Figure 1 is too difficult to read, can be simplified.
  5. Add a figure to show the distribution of the human footprint index will be helpful.
  6. How to deal with those highly correlated variables in the same model?  Please explain.

Author Response

  1. The conclusion in Line 306-307 "Conservation priority of snow leopard is to protect 306 current habitat rather than making movement corridors." require more supporting materials.

Response: This is one of the key points we want to deliver. We added a citation about climate refugia of snow leopards (Li, et al. 2016), which provided an optimistic prediction that the central part of the current snow leopard habitat remained climatically suitability through the late 21st century, supporting viable populations and should function as refugia for snow leopards.

  1. In Line 314 stated "climate change is a potential threat to the snow leopard", but also said in Line 304-305 "We predicted the snow leopard would have slightly larger suitable habitat in the future." This needs to be clarified.

Response: Climate change is a potential threat, and we have two citations here to support this statement. Our research indicated the threat is minor, but we would not totally ignore this threat. The key point of the last paragraph is that we should closely monitor the snow leopard status. We agree there is contradiction here, and this is the status of snow leopard research (some researchers are optimistic and others are not). Anyway, we added “…, although we predicted the threat is minor”.

  1. The discussion on using Random Forest is better than previous modeling reported using other models requires more supports in the discussion.

Response: We added one citation which evaluated the performance of random forest and other models for low density animals in large areas, where random forest was ranked as the best model.

  1. The resolution of all color maps need to be improved.  Also, Figure 1 is too difficult to read, can be simplified.

Response: The low-resolution figures were inserted into the main text for review, and high-resolution images will be provided if the manuscript is accepted. Figure 1 was carefully designed with plenty of information. We intended to demonstrate a sophisticated designed figure rather than a simple one.

  1. Add a figure to show the distribution of the human footprint index will be helpful.

Response: we added one supplementary figure (Fig. S4) showing the human footprint index with snow leopard occurrences.

  1. How to deal with those highly correlated variables in the same model?  Please explain.

Response: A model with highly correlated variables is common in climate change studies, and it is no longer a problem for algorithms such as random forest. Random forest applies tree models to split data based on explanatory variables. Such an algorithm can handle thousands of explanatory variables, and multicollinearity is never a problem for random forest. It uses bootstrap to resample data to generate many trees, and its voting strategy makes the result robust. In the methods section we added “correlated” in the sentence “It is especially good for handling high dimensional data (multiple correlated explanatory variables),…”. We feel we clarified how to deal with highly correlated variables in one model.

Reviewer 2 Report

Thank you for inviting me to be a reviewer of the manuscript entitled Potential range shift of snow leopard in future climate change scenarios. This paper is really impressive in terms of your efforts to demonstrate the power of your model.

I suggest you change the structure of the paper and add a research methodology with research questions and hypotheses. I also suggest adding and clarifying the research objectives. 

The results of your model are amazing and will benefit other scientists. 

Author Response

  • I suggest you change the structure of the paper and add a research methodology with research questions and hypotheses. I also suggest adding and clarifying the research objectives. 

Response: Thanks for providing the comments.

To make our research questions clear, we cited more papers to show current research gaps in the introduction section. Please see “Several studies evaluated the effects of climate change on the snow leopard. Li et al. built the habitat map of the snow leopard from the last glacial maximum to 2070, predicted the climate refugia for this species [8]. Riordan et al. assessed potential connectivity across the range of the snow leopard, suggested corridors for individual dispersal between populations under the pressure of climate change [9]”.

To strengthen our points, we made numerous revision in the last three paragraphs of the manuscript. In the first paragraph, we added: “Li et al. [8] projected that the central part of the current snow leopard habitat remained climatically suitability through the late 21st century, supporting viable populations and should function as refugia for snow leopards. Given the present stable status and future viability at their habitat, we believe…”. In the second paragraph, we cited two papers which saying the snow added declined in some places. Our general view is that the snow leopard population is stable, yet we intend to provide a more realistic status. In the third paragraph, we inserted one clause: “, although we predicted the threat is minor”.

Now we think the research question of this study is clear: project the potential range shift of the snow leopard in the future based on new survey results and latest publications. The hypotheses and research objectives are of the same. We feel we have clarified these fairly enough. To highlight the research gaps, we added “recently obtained” to the sentence “In this paper, we aim to build a range-wide SDM for the snow leopard using recently obtained high quality GPS locations,…”.

  • The results of your model are amazing and will benefit other scientists. 

Response: Thank you very much for the comment.

Reviewer 3 Report

see attached comments provided to the editors

Author Response

Line 24: Would? This is pure speculation! Replace with “may”

Response: Changed as suggested.

28: No snow leopards have been shown to frequently cross 40-100km or wide valleys and high elevation plains, thus not eliminating the possibility of northward shifts, or range adjustments in any of the cardinal compass directions for that matter.

Response: We total agree that snow leopards are not likely to move across wide valleys and high elevation plains, so that we think an upward movement is much easier for snow leopard than northward movement. Even the northward shift could happen, it would not be dramatic. Please see our explanation to “distribution center” in order to clarify the meanings of Figure 6 below (page 7).

46: Author should cite Li et al – among the first papers addressing climate change at the global scale for snow leopards Also see line 55-59. At line 46 we added the citation of Li et al (2016). It is an important work and we should cite it in the first place. The paper by Riordan et al (2016, Ecography:39) is an example of range-wide connectivity study, while that by Li et al. (2016) assessed climate refugia for snow leopards, and should have been briefly reviewed as background to this assessment

Response: We added these two papers as: “Li et al. built the habitat map of the snow leopard from the last glacial maximum to 2070, predicted the climate refugia for this species [8]. Riordan et al. assessed potential connectivity across the range of the snow leopard, suggested corridors for individual dispersal between populations under the pressure of climate change [9].”.

48, 77, : Misleading. There may be considerable error in GPS altitude readings, and can be well in excess of 30 meters for x,y and z parameters, especially in mountainous areas where the number of connections to different satellites is a constraint (the few the number the greater the potential spatial and elevation errors, depending on actual alignment of the detected satellites).

Response: Thanks for pointing out this issue. Here I have a different opinion. I started to use GPS handsets from 1997. Based on my 24-year-long experiences I know the x-y (i.e. Lat/Lon) recoding by a GPS handset is very accurate. The literature indicated the error is about 15 m, and my estimation of the error is 3-5 m. During the surveys, we kept recording all the time, recorded both survey routes and waypoints. Occasionally we can only receive signals from less than four satellites, and the GPS locating algorithm can average the values and provide accurate records. The estimation of altitude by a GPS handset is not accurate, and the error could be a few hundred meters if without a barometer sensor inside the GPS handset. We never used the altitude data from GPS handsets. When we conducted field surveys at the snow leopard habitat, we always have plenty of available satellites since the area is so open.

Also, the authors have not recognized problems related to scale-related mismatching of the different variables. All climate predictions are at a much coarser spatial resolution (i.e., 1km or greater versus the estimated 30m resolution for the other predictor variables). Thus, conclusions reached may be essentially invalidated through incompatible spatial resolutions. This represents a major methodological misunderstanding and potentially significant source for error of this study. Also see comment on the sample size of snow leopard locations used for assessing current and future habitat suitability

Response: All the environmental variables (27 raster layers) have the same spatial resolution of I km. The resolutions of the data are their original resolutions, and we did not change them. The 19 climate variables were spatially interpolated and represent the climate gradient of the study area. Elevation and human footprint index data were generated based on fine scale information. All the environmental variables were well documented and highly cited. We do not think there is mismatch of scales among different variables.

77 Sample selection criteria continued. The authors make questionable assumptions limiting data to 30m resolution (from GPS locations only). In so doing they introduce significant geographic bias, with the model focused heavily on locations in China, specifically the Sanjjiangyuan and Qilianshan reserves, which are certainly not representative of conditions across the species range. Secondly a sample of sone 400 locations is relatively small, also likely to contribute to biased output representing a small subset of habitat conditions (current and forecasted under climate change scenarios). Why not use the same total or adjusted data set employed by Li et al in their assessment of climate refugia, especially as the GPS based locations only capture a small portion of of the study area? See Li, J.; McCarthy, T. M.; Wang, H.; Weckworth, B. V.; Schaller, G. B.; Mishra, C.; Lu, Z.; Beissinger, S. R., Climate refugia of snow leopards in High Asia. Biol. Conserv. 2016, 203, 188-196.

Response: We agree the occurrences used in our study is a biased sample of snow leopard population, and our models focused heavily on the three national parks (Sanjiangyuan, Qilianshan, and Wolong) in China. We also agree that our sample size is small. The snow leopard occurs in remote high mountain areas, and a fully survey of the species is very difficult, and not likely to be completed in future 10 years. As such, what we can do is to use biased sample to represent the distribution of the snow leopard. Fortunately, the random forest algorithm fits such bias data well, which predicts presence for areas with occurrences and leave unsurveyed area to be absent in the model. With the accumulation of survey data, the model would be improved gradually in the future. We didn’t use inaccurate occurrences (e.g. the Lat/Lon values were derived based on the location names, which have errors about several km). As mentioned in the manuscript, in the mountain area a few km distance can result in totally different habitat types.

Li’s data come from a 2008 symposium attended by knowledgeable scientists from almost all of the 12 range countries. The authors could have conducted valid sub-sampling and ended up with a better proportioned sample from across the leopard’s range totaling 4 x the number. Outputs from any modeling exercise is contingent upon the inputs provided, and I find the current dataset woefully lacking in several aspects: first skewed geographic representation, and completely lacking at least one key variable, namely land cover or vegetation site. These deficiencies must be address if the objective is to assess potential climate changes on a range-wide basis.

Response: We agree that Juan Li’s data is stronger than ours. She mixed GPS data and other data from various sources. To be different, we only used GPS data obtained in recent years. As to the skewed sample, please see the previous reply.

As to lacking land cover data, this is a weakness of our study. At present, there is no benchmarking reference for future land cover data. Several datasets are available, but they do not cover all the periods (e.g. 2050, 2070) and all climate change scenarios (RCPs 2.6, 4.5, 6.0, 8.5). To compensate this weakness, we developed a full model using 27 variables. We believe the information of land cover can be largely represented by these 27 variables. The result indicated that the climate model with 18 climate variables had similar performance as the full model. As such, including future land cover data is not highly necessary for this study.

101: The proposed use of the SSP’s could not be implemented. The authors should note this set of variables was replaced with the human footprint index.

Response: Socioeconomic Pathways (SSPs) are proposed to replace or compensate the representative concentration pathways (RCPs). SSPs take into account socioeconomic factors such as population, economic growth, urbanization, education, and the rate of technological development, but it is still different from human footprint index, which is only an environmental variable. We mentioned SSPs in the manuscript because current climate change studies incline to use SSPs rather than RCPs, and we intend to make our position clear.

126: Use of 27 variables is misleading and does not necessarily lead to suitable model. Judicious use of variables should be considered from the very beginning. The variables should reflect important ecological and behavioral drivers for habitat selection and movement by snow leopards. I am puzzled why land-use and vegetation cover types are not include, given their importance to determining habit suitability for this species, as well as its prey items. Simple models, with relatively few well-targeted variables are preferable to throwing everything into the mix, and not removing obviously correlated variables early in the modeling process. What has wind speed got to do with snow leopard ecology for example? i.e the “garbage in, garbage out” adage. Finally, 19 variables is NOT a simple model. Rather, seven variables for the climate model seems more appropriate as the selected model (see line 191)

Response: The current machine learning algorithms such as random forest can take advantage of high dimensional data, and have better performance than simpler models. We used 27 variables in order to maximize the model performance. We explained why we did not use land cover data before. We agree that the prey density is a very important variable, but it is not available. For linear models, a model with a few important variables is better than numerous correlated variables. For machine learning algorithms such as random forest, the logic of model selection has changed. As to the role of wind speed, this variable is associated with microclimate conditions, and is potentially associated with the prey distribution. For random forest, a model with 19 variables is a simple model; it can handle thousands of variables, weight them appropriately, and have good prediction. As such, the number of 19 or 7 does not matter much. We just selected the best model.

The goal should be to select relevant variables that link with snow leopard ecology (including key species habitat preferences) This is not addressed in the present suitability model at all for blue sheep or ibex, the two dominant large ungulate prey species range-wide for SL distribution.

Response: We do not have the data of blue sheep or ibex across the range. It is common that some important variables are missing yet the habitat prediction is still good, as long as the target species is a habitat specialist so that the environmental variables can represent its spatial heterogeneity.

Fig 2, line 198. Again, the current suitability model is very biased toward the Tibet-Qinghai Plateau with the Himalaya under-represented. This presumably reflects the heavy selected of points from this region.

Response: We agree the suitable model is biased towards the three national parks where we have numerous occurrences. Other studies of snow leopard are also biased, since over half of the snow leopard habitat in the world has never been surveyed. Even in Sanjiangyuan National Park, where is extensively surveyed for over 10 years, most mountain ranges have never been reached.

Line 209-214: Addressing potential correlation of variables is helpful. Recommend more discussion in this section as to which variables were considered the most suitable for predicting current habitat suitability.

Response: We added: ”The important variables include temperature and precipitation variables, and the variances of these two factors (e.g. precipitation seasonality, temperature seasonality) were more important than their mean values (e.g. mean temperature, total precipitation). Other variables such as elevation, human footprint index, solar radiation, and water vapor pressure also appeared important.”. Besides, we listed six most important variables in the previous paragraph. We also showed the importance index of all variables in Fig. S2.

216: Use of the mean distribution center simply reflects where most data points are clustered and is thus largely meaningless for distance range portions like the Himalaya, especially given the relatively high mean elevation of the sample data set. The statement of a 100 m elevation increase under the two scenarios is also grossly oversimplified. What about the influence of latitude along with elevation influenced seasonal temperature regimes as these improve or limit primary productivity (i.e., plant growth in turn potential prey suitability under both current and the predicted climate change situation)?

Response: The use of the mean distribution center reflected the range shift trend of all suitable habitat, which was calculated using our species distribution models. Himalaya area was less sampled, so that the predicted range shift has little association with this area. The species distribution model provided a straightforward result: 100 m upward and 200 km northward potential range shift. The question about “the influence of latitude along with elevation influenced seasonal temperature regimes as these improve or limit primary productivity” can be treated by mechanism models. Species distribution models are statistical models, and they never go that deep.

224: The shift in Figure 3 is more toward the northeast, not the NW. Habitat extent vary in the Pamir region depending upon the particular climate change scenario.

Response: We highly appreciate the reviewer looked at the Figure 3 so carefully. There is lot of information in Figure 3. However, finding the range shift (difference between panel A and others) is not easy. After minus calculation (Figure 4) and summarization (Figure 5) we are sure that the snow leopard may have a northwest range shift. We agree that habitat extent vary in the Pamir region depending upon the particular climate change scenarios.

Figure 4 suggests significant declines from climate change for snow leopards with improvement significantly improved in highly non-mountains areas or small mountain patches imbedded with a nonmountainous matrix (e.g., core part of Tibetan Plateau). All suitable areas in future (green areas) are mountainous areas. Please compare Figure 4 with Figure 1. If snow leopards are only expected to move up by an average of 100m elevation, why should the entire Himalayan range be considered less suitable? This area was less sampled, so that it was not predicted suitable. This is a model bias. This makes little sense to me. It is more rational of areas like the Tibet-Qinghai Plateau that already have base altitudes of 4,000m or higher. The snow leopard not only need high elevation, but also need mountainous landform, where its prey occurs.

238: Figure 5 shows habitat shifts falling within area of plains or non-mountainous areas, which are not selected for by this feline. Again, snow leopards are closely tied to mountains and rugged terrain and cannot be expected to occupy or exist in extensive open, flat terrain. Makes no sense to me, so the model is not meeting reasonable predictive threshold targets. Again, the authors must be far more cognizant of the snow leopard’s habitat requirements. The model appears to too gross (or poorly represented) to generate logical habitat and/or geographic projections. They need to start again, perhaps by using the same or similar dataset to that employed by Li et al (2016), and which came from snow leopard experts in the first place. If this produces generally comparable and biologically sensible results using the Random Forest algorithm, all the better.

Response: The distribution center is just the weighted average of values of Lat/Lon in the whole study area (please see lines 223-224 in the original manuscript), so that it is not the occurrence of the snow leopard. The nine points (distribution centers) in Figure 5 were calculated from the nine panels in Figure 3. None of the points was real occurrence. We didn’t predict the snow leopard moving toward plains. We agree that snow leopards are closely tied to mountains and rugged terrain and cannot be expected to occupy or exist in extensive open, flat areas.

246: Figure 6 further highlights the study bias, with the probability distribution concentrated over elevations of 1000 – 1,500 m; these represent a very considerable shift from the species’ current lower elevation limits (except for the Russian or Mongolian populations).

Response: This black curve is about the whole study area, not the snow leopard distribution (please see lines 246-247 in the original manuscript). The red curve is the distribution density of the elevation of snow leopard occurrences. The two curves have nothing to do with climate change, whereas the nine points above are based on climate change analysis.

256: Totally agree that lack of occurrence data is a major constraint. However there are options for addressing this, namely (1) do not limit modeling data to only GPS sources, We tried to use different data and apply different models to generate new results. We set a higher data standard that only uses GPS data, because inaccurate occurrences (e.g. the Lat/Lon values were derived based on the location names, which have errors about several kilometers). As mentioned in the manuscript, in the mountain area a few km distance can result in totally different habitat types. and (2) Follow the Li et al example used for identifying climate refugia by maximizing the dataset for modeling snow leopard habitat suitability; Same as above. and (3) modeling projected vegetation / land cover shifts under the climate change scenarios. I repeated the previous reply here: At present, there is no benchmarking reference for future land cover data. Several datasets are available, but they do not cover all the periods (e.g. 2050, 2070) and all climate change scenarios (RCPs 2.6, 4.5, 6.0, 8.5). To compensate this weakness, we developed a full model using 27 variables. We believe the information of land cover can be largely represented by these 27 variables. The result indicated that the climate model with 18 climate variables had similar performance as the full model. As such, including future land cover data is not highly necessary for this study.

300-317: Comments regarding these conclusions addressed in the above notes.

Response: There are three paragraphs here. In the first paragraph, we added: “Li et al. [8] projected that the central part of the current snow leopard habitat remained climatically suitability through the late 21st century, supporting viable populations and should function as refugia for snow leopards. Given the present stable status and future viability at their habitat, we believe…”. In the second paragraph, we cited two papers which saying the snow added declined in some places. Our general view is that the snow leopard population is stable, yet we intend to provide a more realistic status. In the third paragraph, we inserted one clause: “, although we predicted the threat is minor”.
